| Open Peer Review | Bacteriology | Methods and Protocols

# Performance analysis of the GeneXpert respiratory panel prototype assay for the diagnosis of viral and bacterial upper respiratory tract infections

Linda M. W. Chan,[1] Michael Tang,[2,3] Eddie C. M. Leung,[1] Nicole Y. Y. Lee,[1] Viola C. Y. Chow[1]

**ABSTRACT** GeneXpert respiratory panel (GX-RP) is a new *in vitro* qualitative multiplexed nucleic acid amplification test designed to simultaneously detect and identify 26 common respiratory pathogens from nasopharyngeal swabs (NPSs) in patients exhibiting signs and symptoms of upper respiratory tract infection. This is a retrospective study that compares the prototype assay with two FDA-approved respiratory panels—the BioFire FilmArray (FA) Respiratory 2.1 plus and Hologic Panther Fusion (PF) respiratory assay. A total of 292 NPS specimens from patients with upper respiratory tract infection symptoms were collected from three hospitals in Hong Kong, SAR. There was concordance in 269/292 specimens (92.1%), reaching full agreement with Influenza A, *Bordetella pertussis,* and *Mycoplasma pneumoniae*. Among the discordant specimens, 7 specimens were only positive for GX-RP, and 16 specimens were only positive for their comparators. Codetections were present in 60.9% of the discordant results. GX-RP has an overall positive percent agreement of 93.1%, negative percent agreement of 99.9%, and a prevalence-adjusted bias-adjusted kappa of 99.0%. It showed good agreement when compared with FA respiratory panel or PF respiratory assay.

**IMPORTANCE** Multiplex respiratory pathogen panels are a diagnostic mainstay in patients with upper respiratory tract symptoms. New platforms and improved versions of previous platforms emerge over time. Early evaluation of their diagnostic performance using real-world data is essential for the necessary revisions to be made, which would facilitate public access to improved panels. Our retrospective study provides preliminary evidence that the GeneXpert respiratory panel prototype assay has comparable performance to the BioFire FilmArray and Hologic Panther Fusion respiratory assays and may be an additional candidate in our future toolkit against upper respiratory tract pathogens.

**KEYWORDS** respiratory virus, performance evaluation, nasopharyngeal swabs, respiratory tract infections, Panther Fusion, GeneXpert, FilmArray, multiplex real-time PCR

Upper respiratory infections (URIs) are the leading cause of acute disease globally, accounting for over 40% cases of all global burden of diseases and injuries (1, 2). Although it rarely results in death, the high incidence imposes a significant burden on economic and health care systems (1–3). URIs may also lead to more severe lower respiratory tract infections, which have much higher hospitalization and mortality rates (4). Timely diagnosis using comprehensive respiratory panels allows rapid administration of targeted treatment to facilitate recovery and halt disease transmission (5). This leads to various downstream public health benefits, such as antimicrobial stewardship, reduced length of hospitalization, implementation of appropriate infection control measures, and

Address correspondence to Linda M. W. Chan, clm037@ha.org.hk.

The authors declare no conflict of interest.

enables robust surveillance of pathogen epidemiology nationally and internationally (1, 6). The recent severe acute respiratory syndrome coronavirus 2 (SARS-CoV-2) pandemic has demonstrated the importance of rapid testing in combating URI, and particularly paramount to control epidemics and pandemics (7).

Since respiratory symptoms are non-specific, there is a paradigmatic shift towards syndromic diagnostics where specimens can be tested for multiple respiratory pathogens concurrently. Multiplex real-time polymerase chain reaction (RT-PCR) testing platforms incorporate selected primers of common respiratory pathogens within one reaction tube. It has a fast turnaround time, as fast as 1 h, and the diagnostic yield has greatly increased for viral and atypical bacterial pathogens (8, 9). The recent multicenter RP2+ study showed that a wide range of respiratory viruses were actively circulating during the SARS-CoV-2 pandemic, marking the necessity of syndromic respiratory panels for precise diagnosis (10).

Most URIs are primarily caused by viruses, but can also be caused by bacteria. Common pathogens include influenza A and B (Flu A and B), parainfluenza (PIV), adenovirus (AdV), respiratory syncytial virus (RSV), human metapneumovirus (hMPV), and rhinovirus/enterovirus (RV/EV), although there is an apparent epidemiological shift in URI pathogens before and after the SARS-CoV-2 pandemic (11, 12). The first FDA-approved respiratory panel in 2008 could detect 14 viruses and took 5 h to process (13). Platform performances have improved substantially since then and encompass improvements such as the inclusion of bacterial targets, higher throughput, the ability to distinguish RV/EV, shortened hands-on operator time, and rapid processing time as fast as 1 h (14).

The objective of this study is to examine the diagnostic performance of a prototype version of a new respiratory panel, GeneXpert respiratory panel (GX-RP), by comparing it with two FDA-approved respiratory panels—the BioFire FilmArray (FA) and Hologic Panther Fusion (PF) respiratory panel assays. A comparison of detectable pathogens by them is shown in Table 1.

**TABLE 1** Comparison of detectable analytes by GX-RP, FA RP 2.1 plus, and PF[a,b]

| Assay target | GeneXpert Cepheid Respiratory panel—RUO (GX- RP) | | FA BioFire Respiratory Panel 2.1 plus(FA) | | PF System Hologic AdV/hMPV/RV, Paraflu, SARS-CoV-2/Flu A/B/RSV (PF) | |
|---|---|---|---|---|---|---|
| Virus | | | | | | |
| AdV | A, B, C, D, and E | Ct | A, B, C, D, E, and F | M | A, B, C, D, E, and F | Ct |
| Coronavirus | 229E, HKU1, NL63, and OC43 | Ct | 229E, HKU1, NL63, and OC43 | M | 229E, HKU1, NL63, and OC43 | Ct |
| Flu A | ✓ | Ct | ✓ | M | ✓ | Ct |
| subtype differentiation | H1-pdm09 | M | H1, H3, and H1-pdm09 | M | – | |
| Flu B | ✓ | Ct | ✓ | M | ✓ | Ct |
| hMPV | ✓ | M | ✓ | M | ✓ | Ct |
| PIV | 1, 2, 3, and 4 | Ct | 1, 2, 3, and 4 | M | 1, 2, 3, and 4 | Ct |
| MERS | – | – | ✓ | M | – | – |
| RSV | A and B | Ct | A and B | M | A and B | Ct |
| RV/EV | ✓ | M/Ct | ✓ | M | RV | Ct |
| SARS-COV-2 | ✓ | Ct | ✓ | M | ✓ | Ct |
| Bacteria | | | | | | |
| B. pertussis | ✓ | M | ✓ | M | – | – |
| B. parapertussis | ✓ | M | ✓ | M | – | – |
| C. pneumoniae | ✓ | M | ✓ | M | – | – |
| M. pneumoniae | ✓ | M | ✓ | M | – | – |

[a]AdV = Adenovirus, Flu A = Influenza A H1, H3, and H1 pandemic 2009, Flu B = Influenza B, hMPV = Human metapneumovirus, PIV = Parainfluenza virus 1, 2, 3, and 4, MERS = Middle East respiratory syndrome virus, RSV = Respiratory syncytial virus, RV/EV = Rhinovirus/Enterovirus, SARS-CoV-2 = Severe acute respiratory syndrome coronavirus 2, B. pertussis = Bordetella pertussis, B. parapertussis = Bordetella parapertussis, C. pneumoniae = Chlamydia (Chlamydophila) pneumoniae, M. pneumoniae = Mycoplasma pneumoniae. Analysis method; M = melting curve analysis, Ct = cycle threshold. ✓ = Assay target virus detected.
[b]"–", not applicable.

## RESULTS

### Demographics

A total of 292 nasopharyngeal swab (NPS) specimens were collected and compared using the GX-RP assay against its two comparators—PF/FA. The median age was seven, ranging from 1 month to 97 years old. Specimens from children (age < 18) constituted 65% (191/292) of the overall sample. The sex distribution between females and males was 123/292 (42.1%) and 169/292 (57.9%), respectively.

### GeneXpert respiratory panel testing

The relative prevalence of identified organisms in this study in descending order was RSV (20.5%), PIV (17%), SARS-CoV-2 (14.4%), Flu A (10.9%), RV/EV (10.9%), AdV and Coronavirus (6.9%), Flu B (3.8%), *Mycoplasma pneumoniae* (2.6%), and *Bordetella parapertussis* and *Bordetella pertussis* (<1%). No Flu A-H1-pdm09 strain, Middle East respiratory syndrome virus (MERS), or *Chlamydia pneumoniae* were detected in specimens archived for this study.

A summary of performance characteristics for each individual GX-RP analyte is summarized in Table 2. There were 236 detected organism results with GX-RP and 246 for its comparators, with an overall positive percent agreement (PPA) of 93.1% (229/246). All analytes have a PPA higher than 80% except for *B. parapertussis,* which has a PPA of 66.7% but is limited by a small number of positive cases ($n = 2$). The negative percent agreement (NPA) was above 99% for all except for RSV, which had a NPA of 96.6%. The prevalence-adjusted bias-adjusted kappa (PABAK) was all above 0.97. GX-RP and its comparators showed similar PPA, NPA, PABAK, and McNemar values for each specific analyte. When the analyte results were pooled together, all agreement statistics remained similar, but the McNemar test became statistically significant ($P = 0.041$). Bacterial targets were few in this study ($n \leq 2$), with the exception of *M. pneumoniae* ($n = 6$). All were single analyte detections with no other bacterial targets. Three specimens were implicated in discordant results, and two of the detections were codetected with another virus.

**TABLE 2** Diagnostic performance of GX-RP and comparator assays for target pathogens[a]

| Assay target | No. of tests with results | | | | PPA | NPA | PABAK | McNemar test |
|---|---|---|---|---|---|---|---|---|
| | GX-/C- | GX+/C+ | GX+/C- | GX-/C+ | (95% CI) | (95% CI) | (95% CI) | (p-value) |
| Virus | | | | | | | | |
| AdV | 274 | 14 | 1 | 3 | 0.82 (0.57–0.96) | 1 (0.98–1) | 0.97 (0.93–0.99) | 0.32 |
| Coronavirus | 274 | 16 | 0 | 2 | 0.89 (0.65–0.99) | 1 (0.99–1) | 0.99 (0.95–1) | 0.16 |
| Flu A | 267 | 25 | 0 | 0 | 1 (0.86–1) | 1 (0.99–1) | 1 (0.98–1) | N/A |
| Flu B | 279 | 11 | 0 | 2 | 0.85 (0.55–0.98) | 1 (0.99–1) | 0.99 (0.95–1) | 0.16 |
| hMPV | 281 | 8 | 2 | 1 | 0.89 (0.52–1) | 0.99 (0.98–0.99) | 0.98 (0.94–1) | 0.56 |
| PIV | 251 | 39 | 0 | 2 | 0.95 (0.83–0.99) | 1 (0.99–1) | 0.99 (0.95–1) | 0.16 |
| RSV | 242 | 47 | 1 | 2 | 0.96 (0.86–1) | 0.97 (0.98–1) | 0.98 (0.94–1) | 0.56 |
| RV/EV | 260 | 27 | 2 | 3 | 0.90 (0.74–0.98) | 0.99 (0.98–1) | 0.97 (0.92–0.99) | 0.66 |
| SARS-CoV-2 | 258 | 33 | 0 | 1 | 0.97 (0.85–1) | 1 (99–1) | 0.99 (0.96–1) | 0.32 |
| Bacteria | | | | | | | | |
| *B. pertussis* | 291 | 1 | 0 | 0 | 1 (0.25–1) | 1 (0.99–1) | 1 (0.98–1) | N/A |
| *B. parapertussis* | 288 | 2 | 1 | 1 | 0.67 (0.09–0.99) | 1 (0.98–1) | 0.99 (0.95–1) | 1 |
| *M. pneumoniae* | 286 | 6 | 0 | 0 | 1 (0.54–1) | 1 (0.99–1) | 1 (0.98–1) | N/A |
| Pooled | 3,251 | 229 | 7 | 17 | 0.93 (0.89–0.96) | 1 (0.99–1) | 0.99 (0.98–0.99) | 0.041 |

[a]GeneXpert respiratory panel (GX-RP); Comparators (C) include BioFire FilmArray (FA) plus 2.1 and Hologic Panther Fusion (PF) respiratory panel system. Coronavirus = Coronavirus 229E, NL63, HKU1, and OC43, Influenza A = H1, H3, and H1pdm2009, hMPV = Human metapneumovirus, Parainfluenza virus 1, 2, 3, and 4, RSV = Respiratory syncytial virus, RV/EV = Rhinovirus/Enterovirus, SARS-CoV-2 = Severe acute respiratory syndrome coronavirus 2, *B. pertussis* = *Bordetella pertussis*, *B. parapertussis* = *Bordetella parapertussis*, *M. pneumoniae* = *Mycoplasma pneumoniae*. PPA = percent positive agreement, NPA = negative percent agreement, PABAK = prevalence-adjusted bias-adjusted kappa.

**TABLE 3** Study demographics stratified by specimen concordance and discordance[a]

|  | Age range (median) | Children (<18) no. (%) | Female sex no. (%) |
|---|---|---|---|
| Concordance (n = 269) | 7.5 | 136 (46.6) | 111 (38.0) |
| GX−/C−(n = 72) | 1–97 (80) | 0 (0) | 31 (10.6) |
| GX+/C+ (n = 197) | 1–97 (4) | 136 (46.6) | 80 (27.4) |
| C = FA | 1–17 (6) | 60 | 40 |
| C = PF | 1–12 (9) | 76 | 40 |
| Discordance (n = 23) | 4 | 18 (6.2) | 12 (4.1) |
| GX+/C− (n = 7) | 4 | 6 (2.1) | 4 (1.4) |
| GX−/C+ (n = 16) | 6 | 12 (4.1) | 8 (2.7) |
| C = FA | 1–84 (4) | 11 | 6 |
| C = PF | 5–67 (53) | 1 | 2 |
| Total (n = 292) | 1–97 (7) | 154 (52.7) | 123 (42.1) |

[a]GeneXpert respiratory panel (GX); Comparators (C) include BioFire FilmArray (FA) plus 2.1 and Hologic Panther Fusion (PF) respiratory panel system. Age in years.

## Retrospective analysis of discordant results

Of all the specimens, 269/292 (92.1%) had concordant results between GX-RP and comparator tests, of which 197/292 (67.5%) were consensus positive and 72/292 (24.7%) were consensus negative. 23/292 (7.9%) had discordant results. Among concordant results, children contributed 136/269 (50.6%) of the total specimens and were found positive in all. In contrast, adults contributed to all the negative specimens and 61/197 (31.0%) of the positive specimens. There were fewer females than males (38% vs 62%). Among discordant results, 18/23 (78.3%) were from children, and the sex distributions were similar. Concordant and discordant results stratified by age and sex are shown in Table 3.

Discordant specimens had significantly more codetections (14/23, 60.9%) than concordant specimens (16/197, 8.1%) with an odds ratio of 17.1 (95% CI: 5.9–52.8). The most common respiratory pathogens involved in discordant codetection results are RV/EV (n = 5) and hMPV (n = 3). There were seven specimens with seven pathogens detected only by GX-RP, including hMPV (n = 2), RV/EV (n = 2), RSV (n = 1), AdV (n = 1), and *B. parapertussis* (n = 1). Many of the specimens showed late amplification, suggestive of low analyte levels, ranging from cycle threshold (Ct) values of 35.6 to 44.4. The only exceptions were *B. parapertussis* and a case of RV/EV, which were detected with the melting curve method. According to the manufacturer's guide and further enquiry with Cepheid, the melting curve method would be utilized when RV/EV is not detected by the amplification method, but specific internal algorithms are not elaborated. On the other hand, there were 16 specimens with 17 pathogens detected only by comparators (one specimen with three positive targets), including AdV (n = 3), RV/EV (n = 3), CoV (n = 2), Flu B (n = 2), PIV (n = 2), RSV (n = 2), hMPV (n = 1), SARS-CoV2 (n = 1), and *B. parapertussis* (n = 1). Among these, two AdV demonstrated late amplification with Ct > 36. Other AdVs were detected by melting curve analysis. Details of discordant results can be found in Table 4.

All specimens tested with GX-RP and PF were further analyzed to compare their Ct values (Table 5). Concordant results showed similar Ct values between the two assays, whereas discordant cases had generally higher Ct values than concordant cases, although the sample size is small.

## DISCUSSION

### Comparison of respiratory panels

This study compares GX-RP with either FA or PF. There are no specific respiratory panels that are designated as the gold standard for URI testing. Both FA and PF are FDA-approved for standard laboratory testing. A study that compares PF with an older version of FA (2.0) showed good agreement with a PPA of 96.5% and NPA of 98.4% (15), but a head-to-head comparison of their latest versions is lacking. The two platforms

**TABLE 4** Case information of discordant cases including codetections and Ct values between GX-RP and comparators (FA 2.1 plus or PF)[a,b]

| Case no. | Sex | Age (years) | GX-RP | | FA 2.1 plus | PF | |
| --- | --- | --- | --- | --- | --- | --- | --- |
| | | | Detections | Ct | Detections | Detections | Ct |
| 1 | M | 11 | CoV-OC43 | 27 | CoV-OC43 | | |
| | | | *B. parapertussis* | – | – | | |
| 2 | M | 5 | PIV3 | 25.4 | PIV3 | | |
| | | | RV/EV | – | – | | |
| 3 | F | 18 | PIV4 | 24.6 | PIV4 | | |
| | | | RV/EV | 38.5 | – | | |
| 4 | F | 68 | RSV | 35.6 | – | | |
| 5 | F | 5 | RV/EV | 32 | | RV/EV | 32.7 |
| | | | AdV | 44.4 | | – | – |
| 6 | F | 5 | AdV | 42.5 | | AdV | 36.5 |
| | | | hMPV | 36.5 | | – | – |
| 7 | M | 6 | PIV2 | 28.9 | | PIV2 | 31.9 |
| | | | hMPV | 35.7– | | – | – |
| 8 | F | 3 | – | – | AdV | | |
| 9 | F | 2 | MP | – | MP | | |
| | | | – | | AdV | | |
| 10 | M | 11 | *B. pertussis* | – | *B. pertussis* | | |
| | | | – | – | CoV-HKU1 | | |
| 11 | M | 8 | – | – | CoV-OC43 | | |
| 12 | M | 5 | – | – | Flu B | | |
| 13 | F | 85 | SARS-CoV-2 | 16.8 | SARS-CoV-2 | | |
| | | | – | – | hMPV | | |
| 14 | F | 1 | RV/EV | 35.2 | RV/EV | | |
| | | | – | – | PIV4 | | |
| 15 | M | 2 | Flu A | 24.3 | Flu A | | |
| | | | – | – | RSV | | |
| 16 | F | 3 | CoV-OC43 | 23.3 | CoV-OC43 | | |
| | | | – | – | RSV | | |
| | | | – | – | RV/EV | | |
| 17 | F | 14 | MP | – | MP | | |
| | | | – | – | RV/EV | | |
| 18 | F | 4 | PIV2 | 32.4 | PIV2 | | |
| | | | – | – | RV/EV | | |
| 19 | M | 7 | – | – | *B. parapertussis* | | |
| 20 | F | 45 | – | – | | AdV | 36.8 |
| 21 | F | 61 | – | – | | Flu B | 35.4 |
| 22 | M | 1 | – | – | | PIV3 | 20.6 |
| 23 | F | 68 | – | – | | SARS-CoV-2 | 34.8 |

[a]AdV = Adenovirus, CoV-HKU1 = Coronavirus HKU1, CoV-OC43 = Coronavirus OC43, Flu A = Influenza A, Flu B = Influenza B, hMPV = Human metapneumovirus, PIV2–4 = Parainfluenza virus 2, 3, and 4, RSV = Respiratory syncytial virus, RV/EV = Rhinovirus/Enterovirus, SARS-CoV-2 = Severe acute respiratory syndrome coronavirus 2, *B. pertussis* = *Bordetella pertussis*, *B. parapertussis* = *Bordetella parapertussis*, MP = *Mycoplasma pneumoniae*. Comparator BioFire FilmArray (FA) plus 2.1 does not provide Ct values and therefore omitted.
[b]"–", not applicable.

have historically delivered reliable performance, even if tested after multiple freeze–thaw cycles (9, 16–19). Both have been demonstrated to improve clinical outcomes and yield positive economic impacts (20–23).

Despite the similarities in diagnostic performance, there are key differences that could influence the choice of respiratory panels. First, GX-RP and FA have a processing time of only 45–60 min as opposed to approximately 2.5 h for PF (Table 6). A faster turnaround time would yield greater benefit in high-throughput platforms. Second, each assay covers a different set of pathogens, which makes it more appropriate in

**TABLE 5** Distribution of Ct values for concordant and discordant tests between GeneXpert and comparator (PF)[a,c]

| Assay Target | Concordant | | | Discordant | | | |
|---|---|---|---|---|---|---|---|
| | No. | GeneXpert Ct range (median) | PF Ct range (median) | No. | GeneXpert (GX+/C-) | No. | PF (GX-/C+) |
| AdV | 11 | 26–42.5 (24.1) | 18.3–36.5 (23.3) | 1 | 44.4 | 1 | 36.8 |
| Flu A | 4 | 19.4–29.7 (25.6) | 15.4[b] | – | – | – | – |
| Flu B | 8 | 23.9–38.9 (32.2) | 24–40.2 (29) | – | – | 1 | 35.4 |
| hMPV | 4 | – | 17.8–33.2 (22.8) | 2 | – | – | – |
| PIV | 22 | 19.7–33.2 (27.7) | 17.6–35.8 (27.9) | – | – | 1 | 20.6 |
| RSV | 34 | 18.2–33.3 (22.9) | 18.2–32 (21.4) | 1 | 35.6 | – | – |
| RV/EV | 19 | 19.7–36.9 (26.6) | 17.5–32.7(24.8) | – | – | – | – |
| SARS-CoV-2 | 30 | 14.7–37.8 (23.4) | 15.8–36.4 (22.1) | – | – | 1 | 34.8 |

[a]GeneXpert respiratory panel (GX); Comparators (C) refer to Hologic Panther Fusion (PF) respiratory panel system which provides cycle threshold (Ct) values for viruses presented. Comparator BioFire FilmArray (FA) plus 2.1 does not provide Ct values and therefore omitted. AdV = Adenovirus, Flu A = Influenza A, Flu B = Influenza B, hMPV = Human metapneumovirus, PIV = Parainfluenza virus, RSV = Respiratory syncytial virus, RV/EV = Rhinovirus/Enterovirus, SARS-CoV-2 = Severe acute respiratory syndrome coronavirus 2.
[b]Ct values were not retrievable for three specimens in laboratory record detecting influenza A by comparator.
[c]"–", not applicable.

certain populations depending on the clinical suspicion. GX-RP and FA detect atypical bacteria, which may be more useful in the paediatric population. If MERS is suspected, only FA can reliably detect the pathogen. In a Flu A outbreak, GX-RP and FA allows for subtype differentiation and may improve epidemiological surveillance. PF allows a modular syndromic approach to virus detection but does not detect MERS or bacterial targets (Table 1). Third, PF utilizes real-time amplification for all virus targets, and GX-RP for certain viruses, allowing quantifiable comparisons for Ct values where applicable. It is not known in this study why PA or FA was chosen in each of the cases, but the above reasons are some potential considerations.

## Diagnostic performance

The performance characteristics of GX-RP were similar to FA and PF. Agreement statistics showed NPA, PPA, and PABAK all exceeding 90%. Traditional agreement statistics, such as Kappa, are affected when specimens are dominated by agreement cells. The McNemar test was employed as a more sensitive test to detect differences, which showed a small but statistically significant difference when tests from each analyte were summed. This finding reflects an imbalance between the discordant cells, such that GX-RP is less likely to detect a pathogen (7 vs 17) when it disagrees with its comparators. Nevertheless, it

**TABLE 6** Technical comparison of GX-RP prototype and its comparators (FA and PF)[a]

| | GeneXpert | FA | PF |
|---|---|---|---|
| Manufacturer | Cepheid | bioMérieux | Hologic |
| Assay | Respiratory panel—RUO | Respiratory Panel 2.1 plus | AdV/hMPV/RV Paraflu SARS-CoV-2/Flu A/B/RSV |
| Sample volume | 300 µL | 300 µL | 500 µL |
| Reagent/cartridge storage | 2°C–28°C | 15°C–25°C | 2°C–8°C |
| Principle of detection | Melt curve analysis, amplification curve analysis (Ct) | Melt curve analysis, | Amplification curve analysis (Ct) |
| Hands-on-time | 2 min | 2 min | 5 min |
| Turnaround time | ~1 h | ~45 min | ~2.5 h |
| Throughput | GeneXpert infinity 48 system: 384 tests in 8 h | FA Torch 48 modules: 384 tests in 8 h | Hologic PF system: 300 tests in 8 h |

[a]AdV = Adenovirus, Flu A = Influenza A, FluB = Influenza B, hMPV = Human metapneumovirus, Paraflu = Parainfluenza virus, RSV = Respiratory syncytial virus, RV/EV = Rhinovirus / Enterovirus.

represents only a small number and studies with a larger number of discordant results are needed to better ascertain genuine detection differences across the platforms.

Targets such as Flu A, PIV, RSV, and SARS-CoV-2 showed 95%–100% concordance. For Flu A, false positives are possible if vaccines were used, such as FluMist or other live-attenuated influenza vaccines, but these factors were not explored in our study. As Flu A demonstrated 100% agreement when reported up to group level, further strain-specific accuracy could be explored, especially for reporting imported strains with potential public health threats, such as suspected avian flu strains (H5 and H7).

## Discrepant analysis

Earlier studies have suggested that discordant results could be due to low viral load, specimen degradation, cross-reactivity, reduced amplification efficiency due to competition in specimens with multiple positive targets, or multiple freeze–thaw steps of archived specimen processing (9, 17, 18, 24–26). Clinical history, such as previous infection, could also be important, highlighted by the propensity of AdV and RV/EV to have prolonged residual presence in the upper respiratory tract, causing false positives (27, 28). These are plausible explanations, as most of the discrepant results in our study showed late amplification, raising suspicion that analyte levels are beyond the limit of detection threshold (LoD). We believe codetection may also be a key factor in affecting accuracy, which was present in 14/23 (60.9%) of our discordant specimens (Table 4).

AdV has historically demonstrated low sensitivity for other respiratory panel assays, including BioFire FA version 1.6, with improved sensitivity in subsequent commercial versions by inclusion of more serotypes (17, 18, 29). Our study showed that AdV has one of the lowest PPAs. Possible explanations may include low level of analytes, which may be below the LoD or differences in subtype coverage by the assays, in which the GX-RP detects group A to E, whereas the comparators cover up to group A to F (24, 27). Since subgroups F and G cause gastrointestinal symptoms, and one AdV (Table 4, case 8) not detected by GX-RP had watery diarrhea, a potential explanation would be due to AdV subgroup F not detected by GX-RP but detected by comparators, although this could be over-analysis of one case and beyond the context of respiratory pathogen testing.

## Limitations

There are several limitations due to the retrospective nature of this study. Specimens in this study were freeze–thawed before testing with GX-RP, as compared to fresh specimens subjected to FA or PF. This could potentially affect GX-RP's performance, but our study has found good overall performance. The Ct values comparing GX-RP with PF were similar (Table 5). This is consistent with previous studies that one additional freeze–thaw cycle appears to have a limited impact (24). Testing a fresh specimen is still desirable, and a prospective study design would obviate this issue.

Discrepant results are conventionally resolved by repeat testing or ascertained by other independent methods. This could not be done in this study because of insufficient specimen volume, which could again be alleviated if it was planned in a prospective study. The study is further hampered by the low prevalence of some bacterial targets, such as *B. pertussis and B. parapertussis*, which makes meaningful interpretation difficult. There were no detected *C. pneumoniae* or MERS to allow further analysis.

Additionally, the initial study design does not attempt to account for clinical variables such as age, gender, or co-existing infections that may have a significant impact on performance. Positivity rates are known to be higher in children when compared to adults, but it was striking that specimens from children had a 100% positivity rate in this study. We found most of the discrepant results to occur in children (Table 3). Future studies could explore whether there is differential performance across different subgroups of patients.

## Conclusions

Our study is a preliminary analysis of the diagnostic performance of GX-RP, which shows overall good agreement when compared with FA or PF for rapid diagnoses of URI pathogens. The study is limited by its retrospective study design and small sample size. Future studies could meaningfully inform practice by including larger sample sizes and a prospective study design.

## MATERIALS AND METHODS

### Study design and clinical specimens

This retrospective study was conducted at the Prince of Wales Hospital with specimens collected from three hospitals (Prince of Wales Hospital, North District Hospital, and Alice Ho Miu Ling Nethersole Hospital) in Hong Kong, SAR, between January 2020 and September 2023. NPSs were collected from patients presenting with URI symptoms such as cough, rhinorrhea, and sore throat. They were placed in 3 mL viral transport medium and sent to the laboratory for immediate standard-of-care laboratory testing by FDA-approved commercial assays BioFire FA 2.1 plus respiratory panel (bioMérieux, Durham, NC) or PF respiratory panel (Hologic, San Diego, CA) according to the manufacturer's instructions. Aliquots of these specimens were stored at −70°C after standard testing. Basic demographic information, including age and sex, was collected at the time of specimen collection and retrieved from the digital database retrospectively at the time of study. Specimens were first randomly selected for evaluation with the GX-RP prototype assay (Cepheid, Sunnyvale, CA). For pathogens with low prevalence, such as *B. pertussis*, *B. parapertussis*, and *M. pneumonia,* specimens that tested positive by PF or FA were added to allow better characterization of the PPA. A total of 292 specimens were included. Some pathogens were not available in our locality, such as Flu A H1-pdm 2009 strain, *C. pneumoniae*, and MERS, and could not be included. There was no attempt to further add positive specimens for other purposes such as better distribution of age, sex, or testing by comparator. The selected archived frozen specimens were retrieved and thawed at ambient temperature for testing with GX-RP. Of those selected for this study, 81 had been tested by FA and 211 by PF. The specimens were then subjected to agreement testing. A specimen was defined as consensus positive or negative if all pathogens in the assay had concordant results between GX-RP and the comparator. Discordant specimens were further analyzed for possible explanations, including Ct value (wherever available) and presence of codetections. Traditional discrepant analysis was not performed as there was inadequate volume for many of the samples. The workflow is shown in Fig. 1.

### GX-RP multiplex RT-PCR prototype assay

GX-RP test is an automated *in vitro* diagnostic test for qualitative detection and differentiation for 26 common respiratory pathogens; AdV (A–E), Coronavirus (CoV HKU1, NL63, 229E, and OC43), Flu A with differentiation of H1-pdm 2009 and Flu B, hMPV, PIV 1–4, RSV A and B, RV/EV, SARS-CoV-2, *B. pertussis*, *B. parapertussis*, *M. pneumoniae*, and *C. pneumoniae*. Targets are similar for comparator assays (Table 1). The test is performed on Cepheid GeneXpert Instrument Systems equipped with GeneXpert 10 color modules. The test was performed on retrieved remnant specimens. Approximately 300 µL of specimen was dispensed in the cartridge for automated specimen processing, nucleic acid amplification, and detection of the target sequences in specimens using RT-PCR and melt peak detection. The following pathogens are detected using melt curve analysis: Metapneumovirus, Flu A H1-pdm 2009, *M. pneumoniae*, *C. pneumoniae, B. parapertussis,* and *B. pertussis*. GX-RP requires approximately 1 h/run.

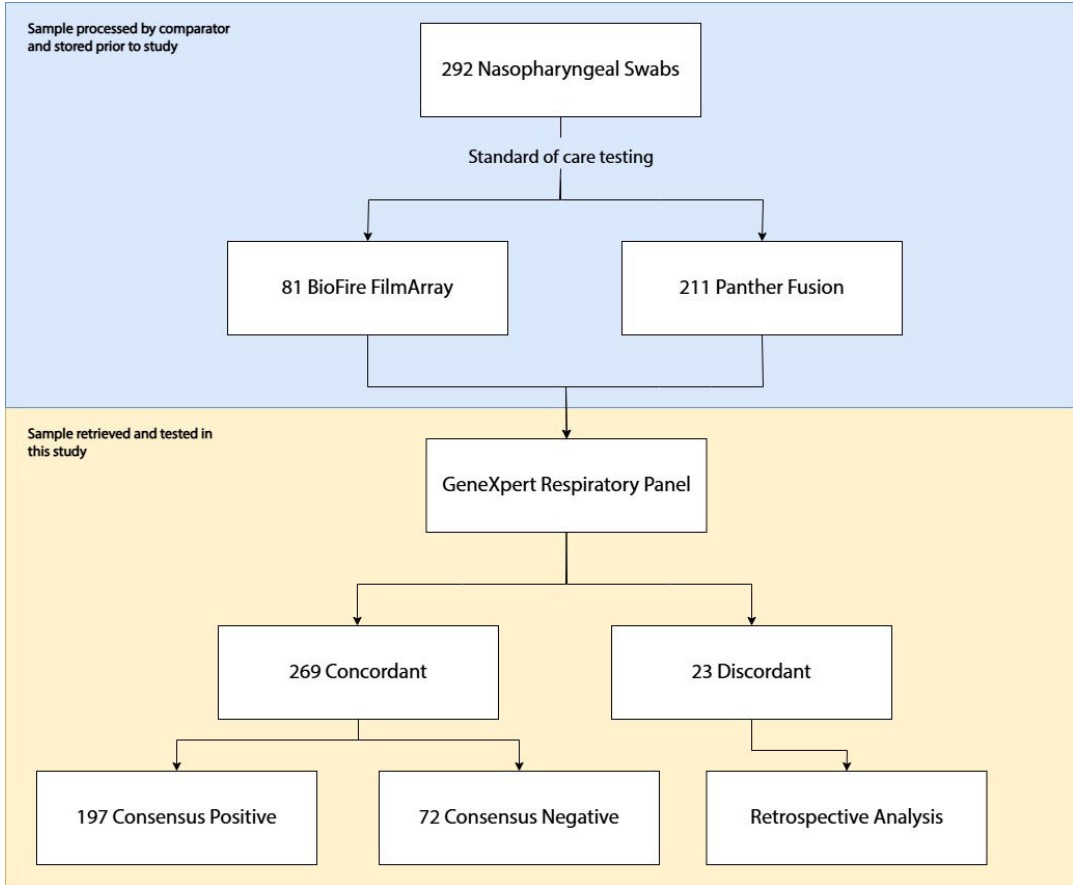

**FIG 1** Study workflow. NPSs were collected from patients with upper respiratory tract symptoms and tested with either BioFire FA or Hologic PF Systems prior to this study. Specimens were then retrieved and tested with GX-RP prototype for performance characteristics. Discordant results were subjected to further analysis.

## Comparator assays

Two FDA-approved commercial assays were used as comparators, which include either FA or PF respiratory panels. The FA respiratory panel is a multiplex nested PCR, performed in a closed and autonomous system, allowing the simultaneous detection of 19 viruses and 4 bacteria (Table 1). A 300 µL of specimen is mixed with specimen buffer and injected into a test pouch containing all necessary reagents for nucleic extraction, amplification, and target detection. The pouch is then placed in the FA instrument. The system software automatically interprets the endpoint melting curve data to provide a qualitative result for each target. A microorganism is reported as detected if at least one of its corresponding assays is positive. The test completion time is approximately 1 h. The PF respiratory panel detects a similar range of viruses as FA without bacterial detection (Table 1). Three different multiplexed RT-PCR tests were used to detect and differentiate respiratory viruses: (i) Flu A/B/RSV; (ii) AdV/hMPV/RV; and (iii) PIV 1–4. The separate panels allow modular syndromic testing. A 500 µL of specimen was transferred to a specimen lysis tube and loaded directly onto the Hologic PF System. This platform performs automated nucleic acid extraction and amplification of the gene target sequences by RT-PCR. The results will be ready in approximately 2.5 h. See Table 6 for comparison of GX-RP and comparator methods.

## Results and statistical analysis

Statistical analyses were performed using R version 4.2.1 with the R packages "epiR" and "fmsb." PF and FA testing were grouped collectively under the term "comparator." The type of agreement statistics included in this study include PPA, NPA, PABAK, and

McNemar's chi-square test. Confidence intervals of 95% for PPA and NPA were constructed using exact intervals. Additional comparative statistics included the Fisher's exact test with a level of statistical significance set at $P$ value of < 0.05.

## ACKNOWLEDGMENTS

The authors appreciate the Microbiology laboratory staff for their relentless support in this study, and also thank Cepheid for providing the equipment, reagents, and technical support for this evaluation.The authors have no other conflict of interest to declare.

## AUTHOR AFFILIATIONS

[1]Department of Microbiology, Prince of Wales Hospital, Hong Kong, Hong Kong SAR
[2]Department of Medicine, Queen Mary Hospital, Hong Kong, Hong Kong SAR
[3]Department of Medicine, Tung Wah Hospital, Hong Kong, Hong Kong SAR

## AUTHOR ORCIDs

Linda M. W. Chan http://orcid.org/0009-0006-2001-5674
Michael Tang http://orcid.org/0000-0001-5381-3003

## AUTHOR CONTRIBUTIONS

Linda M. W. Chan, Conceptualization, Formal analysis, Investigation, Methodology, Project administration, Writing – original draft, Writing – review and editing | Michael Tang, Formal analysis, Methodology, Software, Validation, Writing – original draft, Writing – review and editing | Eddie C. M. Leung, Conceptualization, Data curation, Formal analysis, Investigation, Methodology, Project administration, Resources, Writing – review and editing, Visualization | Nicole Y. Y. Lee, Data curation, Investigation | Viola C. Y. Chow, Resources, Supervision

## ADDITIONAL FILES

The following material is available online.

Open Peer Review

**PEER REVIEW HISTORY (review-history.pdf).** An accounting of the reviewer comments and feedback.

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
