## [Reviewer comments · Microbiology Spectrum]

Microbiology Spectrum

Performance Analysis of the GeneXpert Respiratory Panel Prototype Assay for Diagnosis of Viral and Bacterial Upper Respiratory Tract Infections

Linda Chan, Michael Tang, Eddie Leung, Nicole Lee, and Viola Chow

Corresponding Author(s): Linda Chan, Prince of Wales Hospital

Review Timeline:

Submission Date:	November 18, 2024
Editorial Decision:	April 20, 2025
Revision Received:	June 27, 2025
Accepted:	July 26, 2025

Editor: Michael Owusu

Reviewer(s): The reviewers have opted to remain anonymous.

Transaction Report:

DOI: <https://doi.org/10.1128/spectrum.02560-24>

Re: Spectrum02560-24 (Performance Analysis of the GeneXpert Respiratory Panel Prototype Assay for Diagnosis of Viral and Bacterial Upper Respiratory Tract Infections)

Dear Dr. Linda Chan:

Thank you for the privilege of reviewing your work. Below you will find my comments, instructions from the Spectrum editorial office, and the reviewer comments.

The authors conducted clinical evaluation of the Performance Analysis of the GeneXpert Respiratory Panel Prototype Assay for Diagnosis of Viral and Bacterial Upper Respiratory Tract Infections.

I have reviewed previous review responses from the transferred journal and found most of them addressed in this current version but the aspect on the use of "analyzable" test appear confusing and difficulty to understand. Based on previous authors reviews, I would suggest authors rather focus on the use of pathogens detected per clinical specimen and not "analyzable test".

Again previous comments seem to raise issues about the use of enriched specimen with low prevalent pathogens like Bordetella and Mycoplasma organisms. Traditional evaluation studies usually goes through phases of experiment and clinical/field evaluation. This manuscript appear to describe the real case scenario of using the natural state of clinical specimen to evaluate the performance of assays. The use of enriched samples for selected samples and evaluating this together with other samples seem to create difficulties in understanding the value of this evaluation exercise. If authors would want to include this then it would be good to have a separate section for experimental evaluation where they only use enriched samples for all pathogens under consideration.

Fusing these together make interpretation a bit difficult. Authors could reconsider this in their reviews and respond alongside the current review comments.

Revision Guidelines

Sincerely,
Michael Owusu
Editor
Microbiology Spectrum

Reviewer #1 (Comments for the Author):

Chan et al look to compare the performance analysis of the GeneXpert Respiratory Panel prototype for upper respiratory infections. Overall, this is a sound comparison, showing that GX-RP as compared to the other panels both detected more and less pathogens. The agreement for the treatable conditions is an asset to this study. In the retrospective analysis aspect, there is always some inconsistency since they were not tested ideally under the same conditions and there may have been some nucleic acid degradation. There would be more support to the overall agreement and discussion of the discordant specimens if all specimens underwent this analysis. Then, you can determine if there is greater influence overall rather than just selecting those that were discordant. Consider looking throughout the manuscript for the CoV and either spell or abbreviate consistently. The discussion is sound and essentially supports that this is a reasonable respiratory panel as long as the limitations are understood. It would be helpful in the discussion 4th paragraph to discuss more specifically the Ct values, and consider looking at children and adults separately. MERS is an important analyte on this assay and should be called out, since there are limited options for hospital laboratories currently to detect this. Consider a sensitivity analysis for the discrepant samples in order to determine if there is even a significant change depending on which group they are grouped in for categorization-- the PPA etc may not change regardless. A sensitivity analysis grouped based on which were tested by FA and which by PF may also yield some definition of this new assay, since the input volume is different between the PF and the others-- does that change the sensitivity and does that also need to be called out in this manuscript.

Manuscript title:

Performance Analysis of the GeneXpert Respiratory Panel Prototype Assay for Diagnosis of Viral and Bacterial Upper Respiratory Tract Infections

Submission: Spectrum02560-24

Response to Reviewer #1 Comments

Summary

Thank you very much for taking the time to review this manuscript. Please find the detailed responses below and the corresponding revisions/corrections highlighted in track changes in the re-submitted files.

General reply and comments to authors:

Chan et al look to compare the performance analysis of the GeneXpert Respiratory Panel prototype for upper respiratory infections. Overall, this is a sound comparison, showing that GX-RP as compared to the other panels both detected more and less pathogens. The agreement for the treatable conditions is an asset to this study.

Comment 1: In the retrospective analysis aspect, there is always some inconsistency since they were not tested ideally under the same conditions and there may have been some nucleic acid degradation. There would be more support to the overall agreement and discussion of the discordant specimens if all specimens underwent this analysis. Then, you can determine if there is greater influence overall rather than just selecting those that were discordant.

Thank you for the comment. Due to the retrospective nature of this study, we agree that specimens tested with GX-RP may have sustained greater nucleic acid degradation due to additional freeze-thaw cycles. Previous studies indicate that the current generation of respiratory panels can maintain performance after freeze-thaw cycles to an extent (see reference 9, 15-18, 24). In our study, GX-RP under this supposed disadvantage was still able to fare comparably. We made a crude comparison which showed that the CT values are similar and expect GX-RP, like its comparators, to maintain sensitivity despite an additional freeze-thaw cycle. We have added this as table 5 in our manuscript.

	Concordant			Discordant			
Assay target	#	GeneXpert Ct range (median)	Panther Fusion Ct range (median)	#	GeneXpert (GX+/C-)	#	Panther Fusion (GX-/C+)
Adenovirus	11	26-42.5 (24.1)	18.3-36.5 (23.3)	1	44.4	1	36.8
Influenza A	4	19.4-29.7 (25.6)	15.4*	-	-	-	-
Influenza B	8	23.9-38.9 (32.2)	24-40.2 (29)	-	-	1	35.4

hMPV	4	21.3**	17.8-33.2 (22.8)	2	36.5-35.7(-)	-	-
Parainfluenza virus	22	19.7-33.2 (27.7)	17.6-35.8 (27.9)	-	-	1	20.6
RSV	34	18.2-33.3 (22.9)	18.2-32 (21.4)	1	35.6	-	-
RV/EV	19	19.7-36.9 (26.6)	17.5-32.7(24.8)	-	-	-	-
SARS-COV-2	30	14.7-37.8(23.4)	15.8-36.4(22.1)	-	-	1	34.8

Nevertheless, we agree that a prospective study testing specimens under the same conditions is preferable for greater accuracy. We have also included this in our discussion (**line 187-193**)

Comment 2: Consider looking throughout the manuscript for the CoV and either spell or abbreviate consistently.

Thank you for the feedback. We have standardized all coronavirus to CoV and COVID-19 to SARS-CoV-2 throughout the manuscript.

Comment 3: The discussion is sound and essentially supports that this is a reasonable respiratory panel as long as the limitations are understood. It would be helpful in the discussion 4th paragraph to discuss more specifically the Ct values, and consider looking at children and adults separately.

Thank you for the comment. We have elaborated on the data for discordant specimens in text (line 124-127) and added new tables. Table 3 compares the age and sex for the concordant and discordant tests. Table 5 compares the available CT values sorted by pathogen for concordant and discordant cases.

Comment 4: MERS is an important analyte on this assay and should be called out, since there are limited options for hospital laboratories currently to detect this.

Thank you for the comment. We agree that MERS is an important pathogen especially when it has caused pandemic and have specifically mentioned that only FA can reliably detect it. (line 143-144)

Comment 5: Consider a sensitivity analysis for the discrepant samples in order to determine if there is even a significant change depending on which group they are grouped in for categorization-- the PPA etc may not change regardless. A sensitivity analysis grouped based on which were tested by FA and which by PF may also yield some definition of this new assay, since the input volume is different between the PF and the others-- does that change the sensitivity and does that also need to be called out in this manuscript.

We appreciate the suggestion as this would improve the quality of our study. The sample size is small when it is stratified by each comparator respiratory panel and therefore one

must exert great caution when interpreting. For instance, we attempted a subgroup analysis of our cohort using PF, which showed the following:

Assay target	PPA (95%CI)	NPA (95%CI)	PABAK (95%-CI)	McNemar test (p-value)
Virus				
Adenovirus	91.7 (61.5-99.8)	97.3 (90.7-100)	0.93 (0.81-0.99)	1
Influenza A virus	100 (66.4-100)	100 (95.1-100)	1 (0.91-1)	N/A
Influenza B virus	88.9 (51.5-99.7)	98.6 (92.7-100)	0.95 (0.83-1)	1
hMPV	100 (39.8-100)	97.3 (90.7-99.7)	0.95 (0.82-1)	1
Parainfluenza	95.7 (78.1-99.4)	98.6 (92.7-100)	0.98 (0.9-1)	0.48
RSV	100 (86-99.5)	96.6 (97.8-100)	0.98 (0.94-1)	1
RV/EV	100 (73.5-97.9)	99.2 (97.6-100)	0.97 (0.92-0.99)	N/A
SARS-COV-2	97.1 (83.3-99.9)	100 (95.1-100)	0.98 (0.90-1)	1

The values appear similar to the pooled cohort, but the very wide confidence intervals due to the small sample size was alarming to us. We are compelled to interpret the results as a pooled cohort as the greater sample size makes it a better representation of its diagnostic accuracy with the assumption that neither PF nor FA is better than the others (as discussed in the first paragraph of our discussion). Future studies with greater sample size could compare each of the panel individually.

Response to Editor Comments

Summary

Thank you very much for taking the time to review this manuscript. Please find the detailed responses below and the corresponding revisions/corrections highlighted in track changes in the re-submitted files.

General reply and comments to authors:

Thank you for the privilege of reviewing your work. Below you will find my comments, instructions from the Spectrum editorial office, and the reviewer comments.

The authors conducted clinical evaluation of the Performance Analysis of the GeneXpert Respiratory Panel Prototype Assay for Diagnosis of Viral and Bacterial Upper Respiratory Tract Infections.

Comment 1: I have reviewed previous review responses from the transferred journal and found most of them addressed in this current version but the aspect on the use of "analyzable" test appear confusing and difficulty to understand. Based on previous authors reviews, I would suggest authors rather focus on the use of pathogens detected per clinical specimen and not "analyzable test".

Thank you for the comment. To make things more concise and less confusing, we have removed "analyzable tests" in the abstract and in the results. Things are kept at the specimen or analyte level. We mentioned about pooling the results together once in the results in order for us to discuss the PPA, NPA, and McNemar in its totality, but have otherwise removed them.

Comment 2: Again, previous comments seem to raise issues about the use of enriched specimen with low prevalent pathogens like Bordetella and Mycoplasma organisms. Traditional evaluation studies usually goes through phases of experiment and clinical/field evaluation. This manuscript appear to describe the real case scenario of using the natural state of clinical specimen to evaluate the performance of assays. The use of enriched samples for selected samples and evaluating this together with other samples seem to create difficulties in understanding the value of this evaluation exercise. If authors would want to include this then it would be good to have a separate section for experimental evaluation where they only use enriched samples for all pathogens under consideration. Fusing these together make interpretation a bit difficult. Authors could reconsider this in their reviews and respond alongside the current review comments.

Thank you for the comment and we apologize for the confusion. All specimens were collected as part of real-life clinical evaluation. We removed the term “enrichment” throughout the manuscript as it can lead to confusion to mean “enhancing growth of organisms from culture”. This was not the case, but a matter of adding archived specimens already tested positive with a comparator. This was done after the initial randomization step to increase or “enrich” the prevalence of target pathogen. This is to bolster the sample size to see whether GXP is equally sensitive when compared to FA, essentially allowing us to check the PPA in a real-world scenario. Otherwise, some pathogens would have a sample size of 0 and we cannot calculate the PPA. However, the number of these rare pathogens remained low. The methodology and discussion no longer mention “enrichment” and specifically states the addition of positive specimens to allow calculation of PPA.

Re: Spectrum02560-24R1 (Performance Analysis of the GeneXpert Respiratory Panel Prototype Assay for Diagnosis of Viral and Bacterial Upper Respiratory Tract Infections)

Dear Dr. Linda Chan:

Your manuscript has been accepted, and I am forwarding it to the ASM production staff for publication. Your paper will first be checked to make sure all elements meet the technical requirements. ASM staff will contact you if anything needs to be revised before copyediting and production can begin. Otherwise, you will be notified when your proofs are ready to be viewed.

Sincerely,
Michael Owusu
Editor
Microbiology Spectrum

Reviewer #1 (Comments for the Author):

Revisions in light of the review submitted are acceptable and thoughtful.